# Impact of a surgical ward breakfast buffet on nutritional intake in postoperative patients: A prospective cohort pilot study

Selma C. W. Musters[1]*, Harm H. J. van Noort[2,3], Chris A. Bakker[1], Isabel Degenhart[1], Susan van Dieren[1], Sven J. Geelen[4], Michèle van der Lee[1], Reggie Smith[1], Jolanda M. Maaskant[5], Willem A. Bemelman[1], Els J. M. Nieveen van Dijkum[1], Marc G. Besselink[1], Anne M. Eskes[1,6]*, on behalf of the Amsterdam UMC Peri-operative Surgical Care Group[¶]

1 Department of Surgery, Amsterdam UMC, University of Amsterdam, Cancer Center Amsterdam, Amsterdam, Noord-Holland, The Netherlands, 2 IQ healthcare, Radboud University Medical Centre, Radboud Institute for Health Sciences, Nijmegen, Gelderland, The Netherlands, 3 Department of Surgery, Radboud University Medical Centre, Nijmegen, Gelderland, The Netherlands, 4 Department of Rehabilitation, Amsterdam UMC, University of Amsterdam, Amsterdam Movement Sciences, Amsterdam, Noord-Holland, The Netherlands, 5 Emma Children's Hospital, Amsterdam UMC, University of Amsterdam, Amsterdam, Noord-Holland, The Netherlands, 6 Menzies Health Institute Queensland and School of Nursing and Midwifery, Griffith University, Brisbane, Queensland, Australia

¶ Membership of the Amsterdam UMC Peri-operative Surgical Care Group is listed in the Acknowledgments.
* a.m.eskes@amsterdamUMC.nl (AME); s.c.musters@amsterdamUMC.nl (SCWM)

**Data Availability Statement:** Our dataset contains ethical restrictions for direct public sharing since it involves sensitive human research participant data.

## Abstract

### Background

An early return to normal intake and early mobilization enhances postoperative recovery. However, one out of six surgical patients is undernourished during hospitalization and approximately half of the patients eat 50% or less of the food provided to them. We assessed the use of newly introduced breakfast buffets in two wards for gastrointestinal and oncological surgery and determined the impact on postoperative protein and energy intake.

### Methods

A prospective pilot cohort study was conducted to assess the impact of the introduction of breakfast buffets in two surgical wards. Adult patients had the opportunity to choose between an attractive breakfast buffet and regular bedside breakfast service. Primary outcomes were protein and energy intake during breakfast. We asked patients to report the type of breakfast service and breakfast intake in a diary over a seven-day period. Prognostic factors were used during multivariable regression analysis.

### Results

A total of 77 patients were included. The median percentage of buffet use per patient during the seven-day study period was 50% (IQR 0–83). Mean protein intake was 14.7 g (SD 8.4) and mean energy intake 332.3 kcal (SD 156.9). Predictors for higher protein intake included the use of the breakfast buffet ($\beta = 0.06$, $p = 0.01$) and patient weight ($\beta = 0.13$, $p = 0.01$).

To be more specific, we conducted this study in a group of participants who were treated in one academic hospital, which makes identification of patients possible. We therefore would like to make our data available upon request. Our medical ethical committee can be contacted using the following email addresses: mecamc@amsterdamumc.nl.

**Funding:** This work is supported by an unrestricted innovation grant of the Amsterdam UMC in Amsterdam, the Netherlands. The funder had no role in study design, data collection and analysis, decision to publish, or preparation of the manuscript.

**Competing interests:** The authors have declared that no competing interests exist.

Both use of the breakfast buffet ($\beta$ = 1.00, $p$ = 0.02) and Delirium Observation Scale scores ($\beta$ = -246.29, $p$ = 0.02) were related to higher energy intake.

## Conclusion

Introduction of a breakfast buffet on a surgical ward was associated with higher protein and energy intake and it could be a promising approach to optimizing such intake in surgical patients. Large, prospective and preferably randomized studies should confirm these findings.

## Introduction

The global volume of surgical procedures continues to grow each year [1]. A substantial number of patients undergoing surgery experience postoperative complications [2], which can lead to an increased length of stay, morbidity, and mortality [3]. The risk of postoperative complications can be diminished by improving nutritional and functional status [4]. Therefore, in the postoperative period, early enteral intake and early mobilization should be encouraged, and incorporated in enhanced recovery programs [4].

Despite the emphasis on early nutrition, many surgical patients remain undernourished during hospitalization [5]. Undernutrition is especially common among patients with gastrointestinal conditions [6]. A key element in preventing undernutrition is to optimize the patients' nutritional status, but observational studies have shown that half of the patients eat 50% or less of the food provided to them [7]. Low intake can be due to patient and illness factors but is also known to have other causes, such as poor mealtime environments and frequent mealtime interruptions caused by clinical care [8, 9].

Multiple interventions such as Protected Mealtimes (PMs), room service, and buffet-style service have been introduced to remove these social and environmental barriers [10–12]. Buffet-style interventions offer the opportunity to improve early postoperative mobilization and nudge patients with attractive food displays and tasty food products and also gives patients the opportunity to choose their own meals. Furthermore, interaction and friendliness between patients could provide support.

However, it is unclear whether a breakfast buffet would be used by patients and how it may improve nutritional intake in hospitalized surgical patients compared to usual breakfast services. In this prospective cohort pilot study, we aimed to assess the use of the breakfast buffet during a seven-day study period, and evaluated whether it impacted protein and energy intake in hospitalized patients after mostly gastrointestinal surgery.

## Materials and methods

This study was reported according to applicable criteria of the Strengthening the Reporting of Observational studies in Epidemiology guideline [13]. The Medical Ethics Review Committee of Amsterdam UMC (location: Academic Medical Center, Amsterdam, the Netherlands) concluded that the Medical Research Involving Human Subject Act does not apply to this study (reference number W19_471#19.544). Patients gave verbal and written informed consent to participate in the study.

### Design and setting

Between November 2019 and February 2020, we conducted a prospective cohort pilot study on two wards for patients recovering from oncological-gastrointestinal, oral maxillofacial, and

plastic and reconstructive surgery in a large tertiary referral center. Combined, the two wards had 45 beds.

## Participants

All patients ($\geq$ 18 years) who were able to read and write the Dutch language and who were admitted to the participating wards, were eligible for inclusion. Patients were excluded if they were not allowed to eat (nil-per-mouth) during the entire admission. Patients were also excluded if they were fed by total parenteral nutrition or via nasogastric tube during the entire admission. Patients with isolation precautions (e.g., contact and/or droplet isolation for various types of infections) were excluded from the study.

## Breakfast buffet

The breakfast buffet was initiated as a collaboration between nursing, nutritional, and surgical staff. Two surgeons and a supervisor nurse visited a center in Oslo, Norway to observe its breakfast service. Based on this observation, internal meetings, and financial support in 2019 two central breakfast buffets were created.

Each breakfast buffet included a patient lounge and was staffed by one nutritional care assistant. The breakfast buffet gave patients the opportunity to choose their breakfast at the patient lounge between 8:00 and 8:30 am. The nutritional care assistant advised each patient to make breakfast choices that matched that patient's nutritional status and dietary requirements. Additional food products (e.g., warm crepes, croissants, boiled eggs, and a yogurt bar with toppings) were offered to support the use of the buffet. In addition, to make the buffet attractive, the lounge featured new chairs, tables for two, and some decorative items (e.g., paintings and artificial plants). Soft music was played during breakfast.

Two types of breakfast services were offered to the patients in the surgical wards: the breakfast buffet and the regular breakfast service. Patients were actively invited to make use of the buffet each day, but could also make use of the regular service. Therefore the study consisted out of one group of patients who used the buffet to a greater or lesser extent. The regular breakfast was served at the patient's bedside by a nutritional care assistant. Patients could choose from the regular menu and consume the breakfast in their bed or in room. The regular menu contained bread (whole grain, white and brown), oatmeal porridge, different types of sandwich spreads (e.g., cheese, strawberry jelly, egg salad), seasonings (e.g., pepper, salt, chutney), milk products (e.g., yogurt, custard), fruit (e.g., apples, bananas, oranges), and drinks (e.g., coffee, tea, orange juice, apple juice, lemonade).

## Patient and outcome variables

The outcome of this study is formulated as the protein intake in grams (g) during breakfast, the energy intake in kilocalories (kcal) during breakfast and the use of the buffet during the study period. On the day of admission, each participant received a diary from the nursing staff. The developed diary consisted out of an intake registry form based on the hospital menu (S1 and S2 Files). Patients were instructed to record the different food products and portions they ate for breakfast in their diary and the type of breakfast service they used (the breakfast buffet or regular breakfast service). Gastrointestinal symptoms (e.g., lack of appetite, nausea, full stomach, food tasting different, and difficulty chewing or swallowing) experienced during breakfast were also to be written down in the diary. This diary was tested by two nurses (ID and MvdL) of the participating wards. The nurses evaluated the diary by checking readability, clarity of wording, layout and style. After this evaluation, a minor change was made by adding an example to the diary how to fill in the diary. Data was collected from the first morning after

surgery or first breakfast after admission until the seventh in-hospital day. Two nurses (ID and MvdL) and a researcher (SM) reminded the patients daily to fill in the diaries. At discharge, the diaries were collected.

To reduce reporting bias of protein and energy intake, patients were asked to fill in the diaries directly after breakfast. Reasons for not filling in food intake (e.g. not allowed to eat, ICU admission or discharge within the seven-day period) were collected from patients' medical status. If a patient did not fill in the diary, we checked if the intake was reported that patient's medical status. If so, we reported the intake in the patient's diary afterward.

**Potential prognostic variables.** Potential prognostic factors of nutritional intake were collected from the patients' electronic records, including age [14], sex [15], weight at admission [16], type of admission (elective or unplanned), American Society of Anesthesiologists Physical Status Classification (ASA PS Classification) [16], length of stay (LOS) [17] and type of surgery (colorectal, hepato-pancreato-biliary, esophageal, neuroendocrine, plastic and reconstructive or oral maxillofacial) [18]. Additionally, risk screening scores measured at the day of hospital admission were collected, i.e., the Short Nutritional Assessment Questionnaire (SNAQ) score [19], the Delirium Observation Scale (DOS) score [20], Amsterdam UMC Extension of the Johns Hopkins Highest Level of Mobility scale (AMEXO) score, and the Johns Hopkins Fall risk assessment score. Furthermore, Numeric Rating Scale (NRS) pain scores [21] measured at the day of admission and consequently every morning until the seventh day of admission were retrieved from the electronic patient records. Lastly, gastrointestinal symptoms [18] and having a liquid diet [14, 21] were considered as potential prognostic factors and were collected via the diaries.

## Data analysis

Statistical analyses were performed using the statistical software package R (version 3.6.2). Descriptive statistics were used to summarize patients' baseline characteristics. Continuous variables were presented as mean (M) and standard deviation (SD) or median and interquartile range (IQR) according to the distribution of the variables. Categorical variables were presented as counts and percentages (%).

The total protein and energy intake per breakfast moment was calculated by converting protein and energy composition per 100 g to protein and energy composition per portion (known portion sizes in g). Mean protein and energy intake of the breakfast were calculated per day per patient over the seven-day study period. Consequently, the proportions of the daily protein and energy requirements consumed during breakfast were calculated per patient as a percentage of the daily requirements for which we used the following criteria: the daily protein requirement was 1.2 g per body weight in kilogram (kg) and daily energy requirement was 30 kcal per body weight in kg per day [22]. The percentage of buffet use per patient during the seven-day study period was calculated.

We conducted univariable regression and multivariable linear regression analyses using backward selection. Due to the subject per variable rate in a sample of 77 patients, a maximum of seven variables were selected, which were the ones with the lowest $p$-values in the univariable regression. A two-sided $p$-value $\leq 0.05$ was considered statistically significant. A 95% confidence interval (CI) of the beta coefficient (β) was calculated. Lastly, we calculated the absolute change in protein and energy intake in the multivariable regression analysis when the buffet was used for 100% of the seven-day study period.

**Handling of missing data.** As missing data ($\geq 10\%$) in the dataset occurred the multivariate imputation by chained equations method in R was used [23]. Five independent copies of the data were analyzed. The estimates of the variables were pooled according to Rubin's rules.

The pooled analyses are presented. A complete case analysis was performed as sensitivity analyses [24].

## Results

### Baseline characteristics

A total of 83 patients agreed to participate in the study of whom six patients were excluded from the analysis due to a nil-per-mouth diet during the entire study period and/or absence during the entire study period. Sixty-four patients (83.1%) underwent oncological-gastrointestinal surgery. The median number of gastrointestinal symptoms over the entire study period experienced by patients was 0.7 (IQR 0–1). Over the entire study period, patients experienced a median NRS morning shift pain score of 3 (IQR 2–4). Baseline characteristics are presented in Table 1.

### Use of the breakfast buffet

The total number of patients per day in the cohort varied from 9 to 54 patients because not all patients were allowed to have breakfast each day (Fig 1). The use of the breakfast buffet per day of the study period ranged from 29.8% (14 of 47 eligible patients) on the fourth day and to 50% on the seventh day (12 of 24 eligible patients; Fig 1). The median percentage of buffet use per patient during the seven-day study period was 50% (IQR 0–83; Table 1). Nineteen patients (24.7%) used the breakfast buffet on a daily basis over the entire study period.

**Table 1. Baseline characteristics of the cohort.**

| Variables | (N = 77) | |
|---|---|---|
| Gender, *n* (%) | | |
| Male | 38 | (49.4) |
| Age in years, mean (SD) | 58.2 | (13.9) |
| Length of stay, median (IQR) | 6 | (4–9) |
| Department, *n* (%) | | |
| A | 60 | (77.9) |
| B | 17 | (22.1) |
| Specialism type, *n* (%) | | |
| HPB | 25 | (32.5) |
| Colorectal | 27 | (35.1) |
| Esophageal | 5 | (6.5) |
| Neuroendocrine | 7 | (9.1) |
| Abdominal wall | 5 | (6.5) |
| Reconstructive surgery | 2 | (2.6) |
| OMS | 6 | (7.8) |
| Admission type, *n* (%) | | |
| Elective | 66 | (85.7) |
| Unplanned | 11 | (14.3) |
| Patient undergoing surgery, *n* (%) | | |
| Yes | 65 | (84.4) |
| BMI (kg/m$^2$), mean (SD) | 25.9 | (4.7) |
| Weight (kg), mean (SD) | 80.8 | (17.9) |
| ASA, *n* (%)[a] | | |
| ASA I | 5 | (6.5) |

(*Continued*)

**Table 1.** (Continued)

| Variables | (N = 77) | |
|---|---|---|
| ASA II | 47 | (61.0) |
| ASA III | 13 | (16.9) |
| SNAQ, *n* (%)[b] | | |
| Not at risk | 72 | (93.5) |
| At risk ≥ 3 | 5 | (6.5) |
| JH fall risk, *n* (%) | | |
| Yes | 8 | (10.4) |
| No | 69 | (89.6) |
| AMEXO, median (IQR) | 8 | (3) |
| DOS, median (IQR) | 0 | (0–0) |
| NRS0, median (IQR) | 2 | (3) |
| Percentage use of the breakfast buffet, median (IQR)[c] | 50 | (0–83.3) |
| Number of gastrointestinal symptoms, median (IQR)[d] | 0.7 | (0–1) |
| NRS1–7, mean (IQR)[e] | 3.0 | (2–4) |
| Liquid diet, mean (SD)[f] | 18.8 | (33.8) |

*N*, number of patients; SD, standard deviation; IQR, interquartile range. Specialism type: HPB, hepatic/pancreatic/biliary; OMS, oral maxillofacial surgery; BMI, body mass index; ASA, American Society of Anesthesiologists Physical Status classification; SNAQ, Short Nutritional Assessment Questionnaire; JH fall risk, Johns Hopkins fall risk assessment; AMEXO, Amsterdam UMC Extension of the Johns Hopkins Highest Level of Mobility scale; DOS, Delirium Observation Scale; NRS0, Numeric Rating Scale at baseline (0).

[a] ASA score was not imputed because missing variables only existed for patients not undergoing surgery.

[b] At risk of undernutrition when score ≥ 3.

[c] Percentage of buffet use per patient during the entire study period.

[d] Number of symptoms during the seven-day period (e.g., lack of appetite, nausea, full stomach, food tasting different, difficulty chewing or swallowing).

[e] Median pain score during the study period.

[f] Percentage of days with a liquid diet during the study period.

## Contribution of breakfast buffet to protein and energy intake

During the study period, patients had a mean protein intake during breakfast of 14.7 g (SD 8.4; Table 2). Univariable linear regression analyses showed the seven variables mostly associated with protein intake: use (percentage) of the breakfast buffet over the study period ($\beta = 0.05$, $p \leq 0.01$), weight ($\beta = 0.11$, $p = 0.04$), SNAQ ($\beta = 1.28$, $p = 0.06$), AMEXO ($\beta = 0.90$, $p = 0.04$), percentage of days with a liquid diet during the study period ($\beta = -0.07$, $p = 0.01$), mean NRS pain score during the study period ($\beta = -0.27$, $p = 0.02$) and mean number of gastrointestinal symptoms during the study period ($\beta = -3.3$, $p = 0.02$; Table 3). In the multivariable linear regression, weight ($\beta = 0.13$, $p = 0.01$) and percentage of use of the breakfast buffet ($\beta = 0.06$, $p = 0.01$) were significantly associated with protein intake (Table 4). When patients would have used the buffet during the entire study period, it could have led to a maximum of 6 g higher protein intake.

Mean energy intake during breakfast was 332.3 kcal (SD 156.9; Table 2). For energy intake, two variables showed statistically significant results in the univariable linear regression analyses: use (percentage) of the breakfast buffet during the study period ($\beta = 0.98$, $p = 0.02$) and DOS ($\beta = -241.93$, $p = 0.03$; Table 3). In the multivariable linear regression model, percentage of use of the breakfast buffet ($\beta = 1.00$, $p = 0.02$) and DOS ($\beta = -246.29$, $p = 0.02$) were

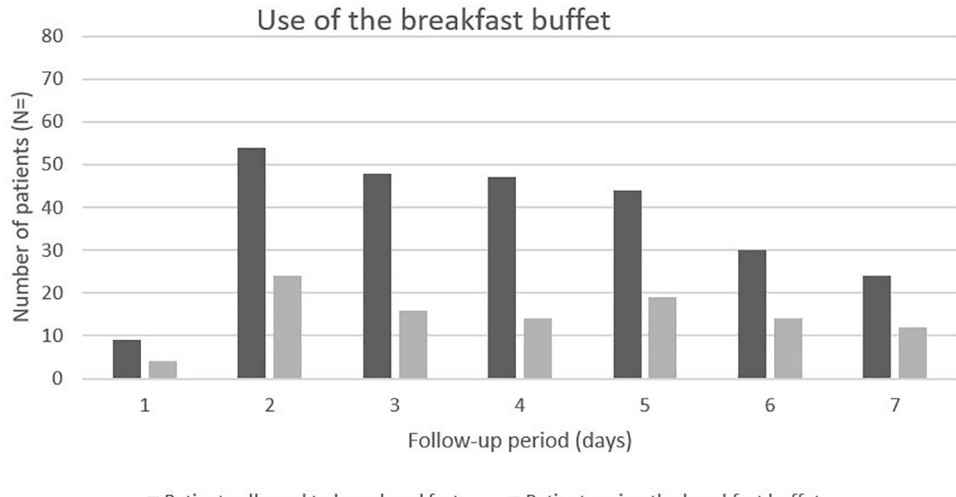

**Fig 1. Use of the breakfast buffet during the study period.**

significantly associated with energy intake (Table 4). When patients would have used the buffet during the entire study period, it could have led to a maximum of 100 kcal higher energy intake.

## Missing data

Missing data of the baseline characteristics complied with missing at random assumptions and ranged from 1.3%–33.1% in the dataset. Complete case analyses showed similar results compared to the pooled analyses (S1–S3 Tables).

## Discussion

This is the first study to assess the use of a central breakfast buffet for surgical patients. We focused on the association of the buffet with protein and energy intake. We found that the median use of the buffet by patients during the study period was 50%, which was significantly associated with higher protein and energy intake.

The breakfast buffet can be considered a complex intervention consisting of a number of interacting components, and requiring new behavior by those delivering (i.e., nurses and nutritional care assistants) and those receiving the intervention (i.e. patients). In this phase of the study, it is not directly possible to draw a straightforward conclusion about which

**Table 2. Protein and energy intake of the cohort during breakfast.**

|  | Protein[a] (g) | | Energy[b] (kcal) | |
|---|---|---|---|---|
| Daily intake, mean (SD)* | 14.7 | (8.4) | 332.3 | (156.9) |
| Percentage of daily requirement (%) | 15.3 | | 14.2 | |
| Estimated daily requirement per day, mean (SD) | 96.5 | (21.3) | 2413.3 | (533.1) |

SD, standard deviation; g, grams; kcal, kilocalories.

[a] Protein intake from breakfast calculated per patient over the seven-day study period.

[b] Energy intake from breakfast calculated per patient over the seven-day study period.

**Table 3. The univariable relationship between potential prognostic factors and protein, energy intake.**

| Potentially prognostic variable | Protein β | SE | 95% CI Lower bound | 95% CI Upper bound | p | Energy β | SE | 95% CI Lower bound | 95% CI Upper bound | p |
|---|---|---|---|---|---|---|---|---|---|---|
| Percentage of use of the breakfast buffet[a] | 0.05 | 1.40 | 0.00 | 0.09 | < 0.01 | 0.98 | 0.42 | 0.13 | 1.83 | 0.02 |
| Age | 0.11 | 0.07 | -0.03 | 0.24 | 0.12 | -0.07 | 1.31 | -2.68 | 2.54 | 0.96 |
| Gender, female | -1.14 | 1.92 | -4.97 | - 2.69 | 0.56 | 3.16 | 36.20 | -68.99 | 75.30 | 0.93 |
| Weight | 0.11 | 0.05 | 0.00 | 0.22 | 0.04 | 0.83 | 1.03 | -1.23 | 2.88 | 0.43 |
| Type of admission, unplanned | 2.60 | 2.74 | -2.85 | - 8.06 | 0.35 | 49.76 | 51.40 | -52.68 | 152.20 | 0.34 |
| ASA | 3.62 | 1.96 | -0.31 | 7.55 | 0.07 | 40.16 | 34.32 | -28.47 | 108.78 | 0.52 |
| Length of stay | -0.04 | 0.17 | -0.39 | 0.30 | 0.80 | -0.60 | 3.22 | -7.00 | 5.81 | 0.85 |
| Surgery specialism | | | | | 0.14 | | | | | 0.58 |
| Abd. wall | 12.64 | 3.66 | 5.33 | 19.94 | <0.01 | 289.12 | 70.30 | 148.84 | 429.41 | <0.01 |
| CR | 1.73 | 3.99 | -6.23 | 9.68 | 0.67 | 41.90 | 76.54 | -110.82 | 194.62 | 0.59 |
| HPB | 3.46 | 4.01 | -4.55 | 11.46 | 0.39 | 73.60 | 77.01 | -80.07 | 227.28 | 0.34 |
| OMS | -4.97 | 4.96 | -14.87 | 4.92 | 0.32 | -31.03 | 95.19 | -220.98 | 158.92 | 0.75 |
| NE | 6.47 | 4.79 | -3.10 | 16.04 | 0.18 | 53.67 | 92.05 | -130.00 | 237.35 | 0.56 |
| OES | -1.46 | 5.18 | -11.79 | 8.88 | 0.78 | -49.28 | 99.42 | -247.67 | 149.11 | 0.62 |
| Rec. | 9.28 | 6.85 | -4.39 | 22.95 | 0.18 | 205.55 | 131.52 | -56.90 | 467.99 | 0.12 |
| SNAQ | 1.28 | 0.68 | -0.08 | - 2.64 | 0.06 | 14.44 | 12.99 | -11.45 | 40.33 | 0.27 |
| DOS | -9.55 | 5.96 | -21.43 | 2.32 | 0.11 | -241.93 | 110.30 | -461.76 | -22.10 | 0.03 |
| AMEXO | 0.90 | 0.43 | 0.03 | 1.78 | 0.04 | 13.10 | 8.57 | -4.47 | 30.95 | 0.14 |
| JH fall risk, no risk | 2.59 | 3.14 | -3.68 | 8.85 | 0.41 | 75.75 | 58.57 | -41.17 | 192.67 | 0.20 |
| Mean NRS pain score[b] | -0.27 | 0.63 | -1.53 | 0.99 | 0.02 | -4.31 | 11.94 | -28.15 | 19.53 | 0.72 |
| Mean number of gastrointestinal symptoms[c] | -3.29 | 1.37 | -6.02 | 0.56 | 0.02 | -40.50 | 26.56 | -93.49 | 12.50 | 0.13 |
| Percentage of having a liquid diet[d] | -0.07 | 0.03 | -0.13 | -0.02 | 0.01 | -0.81 | 0.53 | -1.86 | 0.25 | 0.13 |

β, beta coefficient; CI, confidence interval; SE, standard error; *p*, *p*-value; ASA, American Society of Anesthesiologists Physical Status classification. Specialism: Abd. Wall, abdominal wall surgery; CR, colorectal surgery; HPB, hepatic/pancreatic/biliary surgery; OMS, oral maxillofacial surgery; NE, neuroendocrine surgery; OES, esophageal surgery; Rec., reconstructive surgery; SNAQ, Short Nutritional Assessment Questionnaire; DOS, Delirium Observation Scale; AMEXO, Amsterdam UMC Extension of the Johns Hopkins Highest Level of Mobility scale; JH fall risk, Johns Hopkins fall risk assessment; NRS, Numeric Rating Scale.

[a] Percentage of buffet use per patient during the study period.

[b] Mean pain score during the study period.

[c] Mean number of gastrointestinal symptoms during the seven-day study period (e.g., lack of appetite, nausea, full stomach, food tasting different, difficulty chewing or swallowing).

[d] Percentage of days having a liquid diet over the study period.

**Table 4. The multivariable regression analyses of prognostic factors for protein, energy intake.**

| Prognostic variables | Protein β | SE | 95% CI Lower bound | 95% CI Upper bound | p | Energy β | SE | 95% CI Lower bound | 95% CI Upper bound | p |
|---|---|---|---|---|---|---|---|---|---|---|
| Percentage of use of the breakfast buffet[a] | 0.06 | 0.02 | 0.01 | 0.10 | 0.01 | 1.00 | 0.41 | 0.17 | 1.82 | 0.02 |
| Weight | 0.13 | 0.05 | 0.03 | 0.24 | 0.01 | | | | | |
| DOS | | | | | | -246.29 | 106.94 | -459.48 | -33.10 | 0.02 |
| Multiple linear regression model protein: $R^2$ = 0.13, adjusted $R^2$ = 0.11 | | | | | | | | | | |
| Multiple linear regression model energy: $R^2$ = 0.13, adjusted $R^2$ = 0.11 | | | | | | | | | | |

β, beta coefficient; CI, confidence interval; *p*, *p*-value; DOS, Delirium Observation Scale.

[a] Percentage of use buffet use per patient during the study period.

component works best and which component can explain the association found in our study [25]. In more detail, components from PMs (i.e., mealtime assistance and proper positioning during mealtimes) could have resulted in higher intake [26]. The large scale implemented PMs itself has shown no evidence in improving intake, but the mentioned components of PMs might [26, 27]. Second, we offered the buffet outside the patients' rooms to stimulate early mobilization after surgery. Early mobilization can improve patients' appetites and is strongly recommended by recovery programs [4, 28, 29]. Additionally, eating together and interacting with other patients is known to increase food intake [30]. In our study, the buffet actually became a driver for social interaction between patients on both wards. Therefore the buffet distinguishes itself from other interventions [22]. Combining these aspects may also influence nutritional intake.

Even though the breakfast buffet was associated with higher intake, improving intake in hospitalized patients remains challenging, especially in gastrointestinal patients [22, 31].This was also seen in our study, since we did not achieve the recommended 20%-25% during breakfast of the total daily protein and energy intake requirements (1.2 g/kg/day for protein and 30 kcal/kg/day) [22, 32].

Some challenges need to be addressed when introducing a breakfast buffet.

First, a small investment by the hospital (in our hospital approximately €1700,-) is needed to create and decorate a patient lounge. Second, more major challenges are the logistic aspects (e.g., shifting medical ward round times, shifting morning care by nurses and changing tasks for nutritional care assistants). Therefore, modifying or tailoring the breakfast buffet to varying local contexts in close collaboration with all relevant stakeholders will likely be required.

It should be noted that this study also has some limitations. First, longitudinal data were merged by calculating mean values or percentages over seven days and therefore missing values no longer appeared. On the other hand, if we had decided to impute this data, it probably would have led to unrealistic results as not all patients had the same observational period [33]. Second, patients might not have reported any food intake when feeling too ill or not have reported accurate food portions. Despite this, patient-self report forms to record food intake show acceptable validity [34, 35]. Additionally, we checked medical status when patients did not report any intake. Third, we focused on the association between the buffet and nutritional intake however, in-depth insight in patient experiences and healthcare professional experiences with the buffet is lacking. Collecting qualitative data could have provided valuable insight in practicability and acceptability of the buffet and the way patients experience hospital food and services [34]. It could have also been useful to collect data on healthcare professional experiences since we significantly changed their work environment [35]. Fourth, we did not perform a sample size calculation for this pilot. Results of this study should therefore be interpreted with caution. Even though we did not performed a sample size calculation, we included over 70 patients, which is more than the recommended sample size for a pilot study [36, 37].

Lastly, the breakfast buffet might have been used especially by ambulant, more self-reliant patients who felt less ill, which might be an alternative explanation for the higher protein and energy intake observed in patients who used the breakfast buffet more often. To partly counter this, we operationalized "feeling ill" into a prognostic variable, and our analysis showed no significant contribution between "feeling ill" and protein and energy intake. Although patients were free to choose their type of breakfast service each day, patients who felt too ill might not have profited as much, and, therefore, different interventions might be needed to improve intake in these patients.

A strength of this study is that we were able to provide insight into nutritional intake during breakfast in a seven-day period, which includes the entire hospital admission for most patients. The seven-day period is necessary because patients begin postoperative intake carefully, and

increase the intake according to tolerance over the course of three to four days after surgery [28]. We offered patients who first could only have breakfast in bed or at a small table in their rooms the opportunity to have breakfast in an attractive lounge.

In this pilot cohort study, we cautiously conclude that the use of a breakfast buffet is associated with higher protein and energy intake in patients. The breakfast buffet might be a promising approach in optimizing intake in hospitalized surgical patients. However, we suggest further large-scale prospective, preferably randomized, studies are needed to investigate the effectiveness of each of the components of the buffet and to investigate buffet-style interventions during other meals, on other hospital wards or other hospital settings before it is implemented on large scale. Future research should focus on investigating the difference in nutritional intake between buffet-style interventions and bedside services by executing a cluster-randomized trial. In addition, patients' experiences of buffet-style interventions should be evaluated, as well as healthcare professional experiences of these interventions.

## Supporting information

**S1 Table. Baseline characteristics of the cohort.** Complete case analysis. NA, not applicable; N, number of patients; SD, standard deviation; IQR, interquartile range. Specialism type: HPB, hepatic/pancreatic/biliary; OMS, oral maxillofacial surgery; BMI, body mass index; ASA, American Society of Anesthesiologists Physical Status classification; SNAQ, Short Nutritional Assessment Questionnaire; JH fall risk, Johns Hopkins fall risk assessment; AMEXO, Amsterdam UMC Extension of the Johns Hopkins Highest Level of Mobility scale; DOS, Delirium Observation Scale; NRS0, Numeric Rating Scale at baseline (0).a At risk of undernutrition when score ≥ 3. b Percentage of buffet use per patient during the entire study period. c Number of symptoms during the seven-day study period (e.g. lack of appetite, nausea, full stomach, food tasting different, difficulty chewing or swallowing). d Median pain score during study period. e Percentage of days with a liquid diet during study period.
(DOCX)

**S2 Table. The univariate relationship between potential prognostic factors and protein, energy intake.** Complete case analysis. Complete case analysis. β, beta coefficient; CI, confidence interval; SE, standard error; p, p-value; ASA, American Society of Anesthesiologists Physical Status classification. Specialism: Abd. Wall, abdominal wall surgery; CR, colorectal surgery; HPB, hepatic/pancreatic/biliary surgery; OMS, oral maxillofacial surgery; NE, neuroendocrine surgery; OES, esophageal surgery; Rec., reconstructive surgery; SNAQ, Short Nutritional Assessment Questionnaire; DOS, Delirium Observation Scale; AMEXO, Amsterdam UMC Extension of the Johns Hopkins Highest Level of Mobility scale; JH fall risk, Johns Hopkins fall risk assessment; NRS, Numeric Rating Scale. a Percentage of buffet use per patient during the study period. b Mean pain score during the study period. c Mean number of gastrointestinal symptoms during the seven-day study period (e.g., lack of appetite, nausea, full stomach, food tasting different, difficulty chewing or swallowing). d Percentage of having a liquid diet over the study period.
(DOCX)

**S3 Table. The multivariable regression analyses of prognostic factors for protein, energy intake.** Complete case analysis. Complete case analysis. β, beta coefficient; CI, confidence interval; p, p-value; DOS, Delirium Observation Scale. a Percentage of use buffet use per patient during the study period.
(DOCX)

**S1 File. Patient self-report diary.** Dutch version.
(DOCX)

**S2 File. Patient self-report diary.** English version.
(DOCX)

## Acknowledgments

We thank the members of the nutritional services team and Directorate Services for supporting the design of the central breakfast buffet on both surgical wards, in particular, Ms. H. Ooms, Ms. J. Bakker, Ms. M. van der Kogh and Ms. F. Bakker. We thank dietician and employee of Directorate Services Ms. C. Gouwerok for receiving a database containing the nutritional values of the food products. We thank Dr E. Elder[6] for the English language check of the manuscript. We thank the hepato-pancreato-biliary surgical unit of Oslo University Hospital for demonstrating their breakfast buffet and procedures.

## Author Contributions

**Conceptualization:** Selma C. W. Musters, Harm H. J. van Noort, Chris A. Bakker, Isabel Degenhart, Sven J. Geelen, Michèle van der Lee, Reggie Smith, Marc G. Besselink, Anne M. Eskes.

**Data curation:** Selma C. W. Musters, Harm H. J. van Noort, Jolanda M. Maaskant, Anne M. Eskes.

**Formal analysis:** Selma C. W. Musters, Harm H. J. van Noort, Susan van Dieren, Jolanda M. Maaskant, Anne M. Eskes.

**Investigation:** Selma C. W. Musters, Harm H. J. van Noort, Isabel Degenhart, Michèle van der Lee, Reggie Smith, Anne M. Eskes.

**Methodology:** Selma C. W. Musters, Harm H. J. van Noort, Sven J. Geelen, Jolanda M. Maaskant, Marc G. Besselink, Anne M. Eskes.

**Project administration:** Selma C. W. Musters.

**Resources:** Selma C. W. Musters, Harm H. J. van Noort, Chris A. Bakker, Isabel Degenhart, Michèle van der Lee, Reggie Smith, Jolanda M. Maaskant, Willem A. Bemelman, Els J. M. Nieveen van Dijkum.

**Software:** Selma C. W. Musters, Susan van Dieren, Jolanda M. Maaskant, Anne M. Eskes.

**Supervision:** Susan van Dieren, Jolanda M. Maaskant, Marc G. Besselink, Anne M. Eskes.

**Validation:** Selma C. W. Musters, Jolanda M. Maaskant, Anne M. Eskes.

**Visualization:** Selma C. W. Musters.

**Writing – original draft:** Selma C. W. Musters, Harm H. J. van Noort.

**Writing – review & editing:** Harm H. J. van Noort, Chris A. Bakker, Isabel Degenhart, Susan van Dieren, Sven J. Geelen, Michèle van der Lee, Reggie Smith, Jolanda M. Maaskant, Willem A. Bemelman, Els J. M. Nieveen van Dijkum, Marc G. Besselink, Anne M. Eskes.

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
