## [Decision Letter · Decision Letter 0]

11 Jan 2022

PONE-D-21-10565Impact of a surgical ward breakfast buffet on nutritional intake in postoperative patients: a prospective cohort pilot studyPLOS ONE

Dear Dr. Musters,

Thank you for submitting your manuscript to PLOS ONE. After careful consideration, we feel that it has merit but does not fully meet PLOS ONE’s publication criteria as it currently stands. Therefore, we invite you to submit a revised version of the manuscript that addresses the points raised during the review process.

The manuscript has been evaluated by two reviewers, and their comments are available below.

The reviewers have raised a number of concerns that need attention. They request additional information/changes to the manuscript, including editing for English language, the length of the discussion, the extent to which the conclusions are supported by the results, methodological aspects of the study, and more.

Could you please revise the manuscript to carefully address the concerns raised?

We look forward to receiving your revised manuscript.

Kind regards,

Sebastian Shepherd

Associate Editor

PLOS ONE

Journal Requirements:

2. Please include additional information regarding the survey or questionnaire used in the study and ensure that you have provided sufficient details that others could replicate the analyses. For instance, if you developed a questionnaire as part of this study and it is not under a copyright more restrictive than CC-BY, please include a copy, in both the original language and English, as Supporting Information. If the original language is written in non-Latin characters, for example Amharic, Chinese, or Korean, please use a file format that ensures these characters are visible.

3. Please state whether you validated the questionnaire prior to testing on study participants. Please provide details regarding the validation group within the methods section.

5. One of the noted authors is a group or consortium "Amsterdam UMC Peri-operative Surgical Care Group and the Dutch Science in Surgical Nursing Group." In addition to naming the author group, please list the individual authors and affiliations within this group in the acknowledgments section of your manuscript. Please also indicate clearly a lead author for this group along with a contact email address.

Reviewers' comments:

Reviewer's Responses to Questions

**Comments to the Author**

1. Is the manuscript technically sound, and do the data support the conclusions?

Reviewer #1: Partly

Reviewer #2: Yes

2. Has the statistical analysis been performed appropriately and rigorously? 

Reviewer #1: No

Reviewer #2: Yes

3. Have the authors made all data underlying the findings in their manuscript fully available?

Reviewer #1: Yes

Reviewer #2: No

4. Is the manuscript presented in an intelligible fashion and written in standard English?

Reviewer #1: No

Reviewer #2: Yes

5. Review Comments to the Author

Reviewer #1: Thank you for the opportunity to review this manuscript. I applaud the authors on the concept and their commitment to a patient-centred iniative as well as dedication to the dissemination of findings with this applied research. There is not enough food-service related literature that is published - when so much is happening in practice - and studies such as these advance practice in this area. The authors have also well-established a specific need in the surgical area. The outcome of enhanced energy intake from the buffet is of interest to readers.

I recommend that this article has further refinement to be published in the English Language. Professional grammatical and written support would be beneficial to better showcase the work that has been undertaken.

The methodologies that were employed for this study were reasonable considering the applied nature of the research.

Some of the analyses and conclusions were not clear e.g. The use of 'maximum' to determine success. 'When patients used the buffet over the 229 entire follow-up period, it led to a maximum of 6 g higher protein intake.' I would have loved to have seen some qualitative data and analysis, some of the associational analysis were not very useful in this context and with this sample size.

The publication doesn't seem to follow usual conventions of length/depth and breadth of discussion for an original research article but this is at the discretion of the PLOs One team to determine its suitability.

Reviewer #2: Thank you for the opportunity to review this manuscript. Undernutrition in hopitals is still a matter of concern, even though many effeort have been made to improve the nutritional care. In overall, i think your pilot study is well conducted. There are, however, some concerns. The intervention is multifacetted and that means that it is difficult to draw conclusions. However, I fully understand that you have done more interventions than just/only offering the breakfast menu.

I suggest that you are more cautious in the conclusion as you do not know, how much the interventions concerning the meal environment has been a part of the positive results. Moreover, it also is difficult to draw conclusions drawing on statisics, when this is 'only' a pilot study an no powercalculation is conducted. I think that should be discussed as well.

I think that you are missing some relevant references as there has been made a review of the effect of protected mealtimes:

Porter J, Hanna L. Evidence-Based Analysis of Protected Mealtime Policies on Patient Nutrition and Care. Risk Manag Healthc Policy. 2020;13:713-721. Published 2020 Jul 6. doi:10.2147/RMHP.S224901

Also, I have participated in a ph.d. study on protected mealtimes based on qualitative methodology by Malene Beck, and I think that these studies could contribute to the illumination of among others the patient perspective of hospital meals.

Thus, when planning the next study, I think different ascpets are needed to take into consideration - and this will improve your manuscript if included.

6. PLOS authors have the option to publish the peer review history of their article (what does this mean?). If published, this will include your full peer review and any attached files.

Reviewer #1: No

Reviewer #2: **Yes: **Ingrid Poulsen

---

## [Author Response · Author response to Decision Letter 0]

22 Feb 2022

We have checked the author guidelines again and made some minor adjustments and changed the file naming. We hope the revision meets all style requirements. 

We provided additional information regarding the patient self-report diary, which was used in this pilot study. The additional information can be found in:

- S1 File. Patient self-report diary. Dutch version.

- S2 File. Patient self-report diary. English version.

3. Please state whether you validated the questionnaire prior to testing on study participants. Please provide details regarding the validation group within the methods section.

Thank you for this remark. Based on this comment, we described the patient diary in more detail. Please see also the Supplementary information for the Dutch and English version of this diary. Two nurses tested and provided feedback on the diary used. The nurses evaluated the diary by checking readability, clarity of wording, layout and style. After this evaluation, a minor change was made by adding an example to the diary how to fill in the diary.

We have added this description in the manuscript; see heading Methods, paragraph Patient and outcome variables, line 128-131.

4. We note that you have indicated that data from this study are available upon request. PLOS only allows data to be available upon request if there are legal or ethical restrictions on sharing data publicly. For more information on unacceptable data access restrictions, please see http://journals.plos.org/plosone/s/data-availability#loc-unacceptable-data-access-restrictions

We have carefully read the Data Availability section, paragraph: Human research participant data and other sensitive data) and we fully understand that data sharing contributes to scientific progress. Nevertheless, our dataset contains ethical restrictions for direct public sharing since it involves sensitive human research participant data. To be more specific, we conducted this study in a group of participants who were treated in one academic hospital, which makes identification of patients possible. We therefore would like to make our data available upon request. Our medical ethical committee can be contacted using the following email addresses: mecamc@amsterdamumc.nl.

5. One of the noted authors is a group or consortium "Amsterdam UMC Peri-operative Surgical Care Group and the Dutch Science in Surgical Nursing Group." In addition to naming the author group, please list the individual authors and affiliations within this group in the acknowledgments section of your manuscript. Please also indicate clearly a lead author for this group along with a contact email address

Thank you for this comment. All authors of the Amsterdam UMC Peri-Operative Surgical Care Group have significantly contributed to the study and are therefore all mentioned as co-authors. We have added the contact email address of the lead author (Dr. A.M. Eskes) of this group to the title page, line 26-27.

We have removed the mention of the ‘Dutch Science in Surgical Nursing Group’ since the members of this group who contributed to this manuscript are mentioned as co-authors (AM Eskes, HHJ van Noort, SCW Musters). 

We have included the captions for the Supporting Information at the end of the manuscript, see paragraph Supporting information, line 320-326. 

REVIEWER #1

1. I recommend that this article has further refinement to be published in the English Language. Professional grammatical and written support would be beneficial to better showcase the work that has been undertaken.

We would like to thank the reviewers for this suggestion, and we apologize for the textual errors. Based on this suggestion, we sent our manuscript to a professional editing service (Scribbr) for grammatical and written support. Afterwards, the manuscript was also checked by a native English speaker Dr E. Elder, Menzies Health Institute Queensland and School of Nursing and Midwifery, Griffith University, Brisbane, Queensland, Australia (see Acknowledgements, line 330). We hope this version meets your expectation.

2. Some of the analyses and conclusions were not clear e.g. The use of 'maximum' to determine success. 'When patients used the buffet over the 229 entire follow-up period, it led to a maximum of 6 g higher protein intake.' I would have loved to have seen some qualitative data and analysis, some of the associational analysis were not very useful in this context and with this sample size.

Thank you for this critical remark. We strived to give a meaningful translation of the beta coefficients to the clinical practice. Since this study is one-armed, we did not have the opportunity to compare protein and energy intake between study arms (e.g. use of breakfast buffet versus use of a regular breakfast service). However, we understand that the use of ‘maximum’ in this sample size does not provide clarification. We hope the following changes in the manuscript provide a more useful and careful description: 

- Results, Contribution of breakfast buffet to protein and energy intake, line 226-227: “When patients would have used the buffet during the entire study period, it could have led to a 6 g higher protein intake.”

- Results, Contribution of breakfast buffet to protein and energy intake, line 234-235: “When patients would have the buffet during the entire study period, it could have led to a 100 kcal higher energy intake.”

We also like to thank the reviewer for the suggestion to collect qualitative data on the use of the breakfast buffet. We agree with the reviewer that it would provide meaningful insight in patients’ experiences and healthcare caregivers’ experiences of the breakfast buffet. In this phase of investigating the use of the breakfast buffet, we first sought to investigate how it could influence protein and energy intake in patients. We have now added this limitation to our discussion section, line 287-292. In future research, we will address collecting qualitative data and we attached this to our discussion section as well, line 316-318.

3. The publication doesn't seem to follow usual conventions of length/depth and breadth of discussion for an original research article but this is at the discretion of the PLOs One team to determine its suitability.

We have now made some changes in the discussion based on both reviewers’ comments. 

REVIEWER #2

1. There are, however, some concerns. The intervention is multifaceted and that means that it is difficult to draw conclusions. However, I fully understand that you have done more interventions than just/only offering the breakfast menu.

I suggest that you are more cautious in the conclusion as you do not know, how much the interventions concerning the meal environment has been a part of the positive results.

Thank you for this comment. We do agree that it is a multifaceted intervention as the breakfast buffet can be considered as a complex intervention whereby a complex intervention is defined as “an intervention with a number of interacting components which requires new behaviours by those delivering and receiving the intervention [25].” We incorporated this information in the discussion section (Discussion, line 258-265) and we formulated our conclusion more cautiously (Discussion, line 307-309). 

2. Moreover, it also is difficult to draw conclusions drawing on statistics, when this is 'only' a pilot study and no powercalculation is conducted. I think that should be discussed as well.

Thank you for reminding us of this point. We understand that the results of this study should be interpreted with caution, since a sample size calculation is lacking. We therefore added the following to the limitation section of the discussion:

- Discussion, line 293-295: “Fourth, we did not perform a sample size calculation for this pilot. Results of this study should therefore be interpreted with caution. Even though we did not performed a sample size calculation, we included over 70 patients, which is more than the recommended sample size for a pilot study [36, 37].”

Furthermore, we have also changed our conclusion:

- Discussion, line 309-311: “In this pilot cohort study, we cautiously conclude that the use of a breakfast buffet is associated with higher protein and energy intake in patients. The breakfast buffet might be a promising approach in optimizing intake in hospitalized surgical patients.”

- Abstract, line 51-53: “Introduction of a breakfast buffet on a surgical ward was associated with higher protein and energy intake and could be a promising approach to optimize intake in surgical patients. Large, prospective and preferably randomized studies should confirm these findings.”

3. I think that you are missing some relevant references as there has been made a review of the effect of protected mealtimes.

We would like to thank the reviewer for these interesting references. We have carefully read the articles [26, 34, 35] and found them indeed relevant for our discussion section. We included important information of the suggested references into the discussion section.

First, we changed the following section, since no evidence was found on PM improving nutritional intake [26]:

- Discussion, line 262-265: “In more detail, components from PMs (i.e., mealtime assistance and proper positioning during mealtimes) could have resulted in higher intake [26]. The large scale implemented PMs itself has shown no evidence in improving intake, but the mentioned components of PMs might [26, 27].

Second, the importance of collecting data on experiences of patients and healthcare professionals with foodservice interventions was explained [34, 35] and we have therefore added the following statement in the limitation section of the discussion:

- Discussion, line 287-292: “Third, we focused on the association between the buffet and nutritional intake however, in-depth insight in patient experiences and healthcare professional experiences with the buffet is lacking. Collecting qualitative data could have provided valuable insight in practicability, acceptability and the way patients experience hospital food and services [34]. It could have also been useful to collect data on healthcare professional experiences since we significantly changed their work environment [35].

Third, in the suggested reference [26] we have read that PM was quickly implemented on international level with lack of evidence for improving nutritional intake. Based on this given, we felt we should be more careful with our conclusion and therefore changed it:

- Conclusion, line 311-318: “… However, we suggest further large-scale prospective, preferably randomized, studies are needed to investigate the effectiveness of each of the components of the buffet and to investigate buffet-style interventions during other meals, on other hospital wards or other hospital settings before it is implemented on large scale. Future research should focus on investigating the difference in nutritional intake between buffet-style interventions and bedside services by executing a cluster-randomized trial. In addition, patients’ experiences of buffet-style interventions should be evaluated, as well as healthcare professional experiences of these interventions.”

---

## [Editor Report · Decision Letter 1]

4 Apr 2022

Impact of a surgical ward breakfast buffet on nutritional intake in postoperative patients: a prospective cohort pilot study

PONE-D-21-10565R1

Dear Dr. Musters, 

We’re pleased to inform you that your manuscript has been judged scientifically suitable for publication and will be formally accepted for publication once it meets all outstanding technical requirements.

Kind regards,

Ingrid Poulsen

Guest Editor

PLOS ONE

Additional Editor Comments (optional):

Dear authors,

I think that you have revised the manuscript carefully and in order to the comments from reviewers and editor.
---

## [Editor Report · Acceptance letter]

19 Apr 2022

PONE-D-21-10565R1 

Impact of a surgical ward breakfast buffet on nutritional intake in postoperative patients: a prospective cohort pilot study 

Dear Dr. Musters:

I'm pleased to inform you that your manuscript has been deemed suitable for publication in PLOS ONE. Congratulations! Your manuscript is now with our production department. 

Kind regards, 

on behalf of

Dr. Ingrid Poulsen 

Guest Editor

PLOS ONE